# Recurrent Flow Networks:
# A Recurrent Latent Variable Model for Density Modelling of Urban Mobility

Daniele Gammelli [1]   Filipe Rodrigues [1]

## Abstract

Mobility-on-demand (MoD) systems represent a rapidly developing mode of transportation wherein travel requests are dynamically handled by a coordinated fleet of vehicles. Crucially, the efficiency of an MoD system highly depends on how well supply and demand distributions are aligned in spatio-temporal space (i.e., to satisfy user demand, cars have to be available in the correct place and at the desired time). When modelling urban mobility as temporal sequences, current approaches typically rely on either (i) a spatial discretization (e.g. ConvLSTMs), or (ii) a Gaussian mixture model to describe the conditional output distribution. In this paper, we argue that both of these approaches could exhibit structural limitations when faced with highly complex data distributions such as for urban mobility densities. To address this issue, we introduce *recurrent flow networks* which combine deterministic and stochastic recurrent hidden states with conditional normalizing flows and show how the added flexibility allows our model to generate distributions matching potentially complex urban topologies.

## 1. Introduction

With the growing prevalence of smart mobile phones in our daily lives, companies such as Uber, Lyft, and DiDi have been pioneering Mobility-on-Demand (MoD) and online ride-hailing platforms as a solution capable of providing a more efficient and personalized transportation service. Notably, an efficient MoD system could allow for reduced idle times and higher fulfillment rates, thus offering a better user experience for both driver and passenger groups. The efficiency of an MoD system highly depends on the ability to model and accurately forecast the need for transportation, such to enable service providers to take operational deci-

---
[1]Department of Technology, Management and Economics, Technical University of Denmark, Denmark. Correspondence to: Daniele Gammelli <daga@dtu.dk>.

Third workshop on *Invertible Neural Networks, Normalizing Flows, and Explicit Likelihood Models* (ICML 2021). Copyright 2021 by the author(s).

sions in strong accordance with user needs and preferences. However, the complexity of the geo-spatial distributions characterizing MoD demand requires flexible models that can capture rich, time-dependent 2d patterns and adapt to complex urban geographies (e.g. presence of rivers, irregular landforms, etc.).

Historically, dynamic Bayesian networks (DBNs), such as hidden Markov models (HMMs) and state space models (SSMs) (Durbin & Koopman, 2001), have characterized a unifying probabilistic framework with illustrious successes in modelling time-dependent dynamics. Advances in deep learning architectures however, shifted this supremacy towards the field of Recurrent Neural Networks (RNNs). At a high level, both DBNs and RNNs can be framed as parametrizations of two core components: 1) a *transition* function characterizing the time-dependent evolution of a learned internal representation, and 2) an *emission* function denoting a mapping from representation space to observation space. Recently, evidence has been gathered in favor of combinations bringing together the representative power of RNNs with the consistent handling of uncertainties given by probabilistic approaches (Chung et al., 2015; Fraccaro et al., 2016; Krishnan et al., 2016; Karl et al., 2017). The core concept underlying recent developments is the idea that, in current RNNs, the only source of variability is found in the conditional emission distribution (i.e. typically a unimodal distribution or a mixture of unimodal distributions). Most efforts have therefore concentrated in building models capable of effectively propagating uncertainty in the transition function of RNNs.

In this paper, we build on these recent advances by shifting the focus towards more flexible emission functions. We suggest that the traditional treatment of output variability through the parametrization of either (i) unimodal (or mixtures of unimodal) distributions, or (ii) discretized representations of naturally-continuous distributions, may act as a bottleneck in cases characterized by complex data distributions, such as the ones observed in urban mobility. We propose the use of Conditional Normalizing Flows (CNFs) (Winkler et al., 2020) as a general approach to define arbitrarily expressive output probability distributions under temporal dynamics. On one hand, we model the temporal variability in the data through a transition function combining stochastic and deterministic states, on the other, we

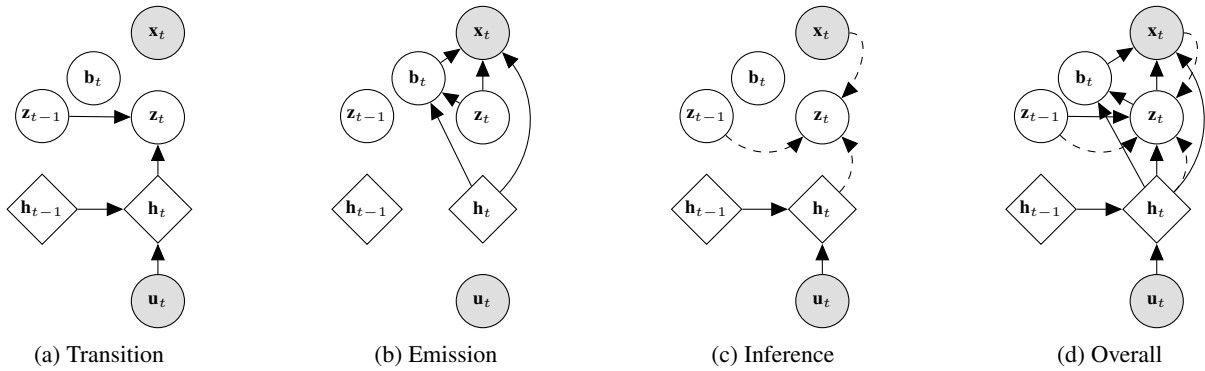

(a) Transition       (b) Emission       (c) Inference       (d) Overall

*Figure 1.* Graphical model of the operations defining the RFN: a) transition function defined in Eq. (1) and Eq. (2); b) emission function as in Eq. (3) and Eq. (4); c) inference network using Eq. (5); d) overall RFN graphical model. Shaded nodes represent observed variables, while un-shaded nodes represent either deterministic (diamond-shaped) or stochastic (circles) hidden states. For sequence generation, a traditional approach is to use $\mathbf{u}_t = \mathbf{x}_{t-1}$.

propose to use this mixed hidden representation as a conditioning variable to capture the output variability with a CNF. We call this model a *Recurrent Flow Network* (RFN).

To summarize, the main contributions of this paper are twofold: first, we propose a probabilistic neural generative model which is able to combine deterministic and stochastic temporal representations with the flexibility of normalizing flows in the conditional output distribution. Second, we showcase how our model is able to represent fine-grained urban mobility patterns on several real-world tasks, which could drastically impact downstream decision making processes in current mobility systems.

## 2. Recurrent Flow Networks

In this section, we define the generative model $p_\theta$ and inference network $q_\phi$ characterizing the RFN for the purpose of sequence modelling[1]. RFNs explicitly model temporal dependencies by combining deterministic and stochastic layers. The resulting intractability of the posterior distribution over the latent states $\mathbf{z}_{1:T}$, as in the case of VAEs (Kingma & Welling, 2014; Rezende et al., 2014), is further approached by learning a tractable approximation through *amortized variational inference*. The schematic view of the RFN is shown in Fig 1.

**Generative model** As in the case of the SRNN (Fraccaro et al., 2016), the transition function of the RFN interlocks an SSM with an RNN:

$$\mathbf{h}_t = f_{\theta_{\mathbf{h}}}(\mathbf{h}_{t-1}, \varphi_\tau^{\text{extr}}(\mathbf{u}_t)) \tag{1}$$

$$\mathbf{z}_t \sim \mathcal{N}(\boldsymbol{\mu}_{0,t}, \text{diag}(\boldsymbol{\sigma}_{0,t}^2)), \tag{2}$$
$$\text{with } [\boldsymbol{\mu}_{0,t}, \boldsymbol{\sigma}_{0,t}] = f_{\theta_{\mathbf{z}}}(\mathbf{z}_{t-1}, \mathbf{h}_t),$$

where $\boldsymbol{\mu}_{0,t}$ and $\boldsymbol{\sigma}_{0,t}$ represent the parameters of the conditional prior distribution over the stochastic hidden states

---

[1]Code available at https://github.com/DanieleGammelli/recurrent-flow-nets

$\mathbf{z}_{1:T}$. In our implementation, $f_{\theta_{\mathbf{h}}}$ and $f_{\theta_{\mathbf{z}}}$ are respectively an LSTM cell and a deep feed-forward neural network, with parameters $\theta_{\mathbf{h}}$ and $\theta_{\mathbf{z}}$. In Eq. (1), $\varphi_\tau^{\text{extr}}$ can also be a neural network extracting features from $\mathbf{u}_t$. Unlike the SRNN, the learned representations (i.e. $\mathbf{z}_{1:T}$, $\mathbf{h}_{1:T}$) are used as conditioners for a CNF parametrizing the output distribution. That is, for every time-step $t$, we learn a complex distribution $p(\mathbf{x}_t|\mathbf{z}_t, \mathbf{h}_t)$ by defining the conditional base distribution $p(\mathbf{b}_t|\mathbf{z}_t, \mathbf{h}_t)$ and conditional coupling layers (Dinh et al., 2017) for the transformation $T_\psi$ as follows:

Conditional Prior:

$$\mathbf{b}_t \sim \mathcal{N}(\boldsymbol{\mu}_{b,t}, \text{diag}(\boldsymbol{\sigma}_{b,t}^2)), \tag{3}$$
$$\text{with } [\boldsymbol{\mu}_{b,t}, \boldsymbol{\sigma}_{b,t}] = f_\psi(\mathbf{z}_t, \mathbf{h}_t)$$

Conditional Coupling:

$$\mathbf{b}_{t,d+1:D} = \mathbf{x}_{t,d+1:D} \odot \exp\left(s_\psi(\mathbf{x}_{t,1:d}, \mathbf{z}_t, \mathbf{h}_t)\right) + $$
$$+ t_\psi(\mathbf{x}_{t,1:d}, \mathbf{z}_t, \mathbf{h}_t) \tag{4}$$
$$\mathbf{b}_{t,1:d} = \mathbf{x}_{t,1:d},$$

In Eq. 4 we assume vectors $\mathbf{x}$ and $\mathbf{b}$ to be $D$-dimensional vectors with $d < D$. In our implementation, $f_\psi$, $s_\psi$ and $t_\psi$ are parametrized by deep neural networks. Together, Eq. (3) and Eq. (4) define the emission function, enabling the generative model to result in the factorization $p(\mathbf{x}, \mathbf{z}, \mathbf{h}) = \prod_{t=1}^{T} p_{\theta_{\mathbf{x}}}(\mathbf{x}_t|\mathbf{z}_t, \mathbf{h}_t) p_{\theta_{\mathbf{z}}}(\mathbf{z}_t|\mathbf{z}_{t-1}, \mathbf{h}_t) p_{\theta_{\mathbf{h}}}(\mathbf{h}_t|\mathbf{h}_{t-1}, \mathbf{u}_t)$, where the emission and transition distributions have parameters $\theta_{\mathbf{x}}, \theta_{\mathbf{z}}, \theta_{\mathbf{h}}$, and where we assume that $\mathbf{h}_t$ follows a delta distribution centered in $\mathbf{h}_t = f_{\theta_{\mathbf{h}}}(\mathbf{h}_{t-1}, \mathbf{u}_t)$.

**Inference** The variational approximation defining the RFN directly depends on $\mathbf{z}_{t-1}$, $\mathbf{h}_t$ and $\mathbf{x}_t$ as follows:

$$q_\phi(\mathbf{z}_t|\mathbf{z}_{t-1}, \mathbf{h}_t, \mathbf{x}_t) = \mathcal{N}(\boldsymbol{\mu}_{z,t}, \text{diag}(\boldsymbol{\sigma}_{z,t}^2)), \tag{5}$$
$$\text{with } [\boldsymbol{\mu}_{z,t}, \boldsymbol{\sigma}_{z,t}] = \varphi_\tau^{\text{enc}}(\mathbf{z}_{t-1}, \mathbf{h}_t, \mathbf{x}_t),$$

where $\varphi_\tau^{\text{enc}}$ is an encoder network defining the parameters of the approximate posterior distribution $\boldsymbol{\mu}_{z,t}$ and $\boldsymbol{\sigma}_{z,t}$. Given

the above structure, the generative and inference models are tied through the RNN hidden state $\mathbf{h}_t$, resulting in the factorization given by:

$$q_\phi(\mathbf{z}_{1:T}|\mathbf{x}_{1:T}) = \prod_{t=1}^{T} q_\phi(\mathbf{z}_t|\mathbf{z}_{t-1}, \mathbf{h}_t, \mathbf{x}_t). \qquad (6)$$

In addition to the explicit dependence of the approximate posterior on $\mathbf{x}_t$ and $\mathbf{h}_t$, the inference network defined in Eq. (5) also exhibits an implicit dependence on $\mathbf{x}_{1:t}$ and $\mathbf{h}_{1:t}$ through $\mathbf{z}_{t-1}$. This implicit dependency on all information from the past can be considered as resembling a *filtering* approach from the state-space model literature (Durbin & Koopman, 2001). Denoting $\theta$ and $\phi$ as the set of model and variational parameters respectively, variational inference offers a scheme for jointly optimizing parameters $\theta$ and computing an approximation to the posterior distribution by maximizing the *evidence lower bound*[2] (i.e. ELBO).

## 3. Experiments

Concretely, we evaluate the proposed RFN on three transportation datasets:

**NYC Taxi (NYC-P/D)**: This dataset is released by the New York City Taxi and Limousine Commission. We focused on aggregating the taxi demand in 2-hour bins for the month of March 2016 containing 249,637 trip geo-coordinates. We further differentiated the task of modelling pick-ups (i.e. where the demand is) and drop-offs (i.e. where people want to go). In what follows, we denote the two datasets as NYC-P and NYC-D respectively.

**Copenhagen Bike-Share (CPH-BS)**: This dataset contains geo-coordinates from users accessing the smartphone app of Donkey Republic, one of the major bike sharing services in Copenhagen, Denmark. As for the case of New York, we aggregated the geo-coordinates in 2-hour bins for the month of August, resulting in 87,740 app accesses.

**Models** We compare the proposed RFN against various baselines assuming both continuous and discrete support for the output distribution. In particular, in the continuous case (i.e. where we assume to be modelling a 2-dimensional distribution directly in longitude-latitude space), we consider RNN, VRNN (Chung et al., 2015) and SRNN (Fraccaro et al., 2016) models each using two different emission distributions based on Mixture Density Networks (MDN), as in (Bishop, 1994). That is, we compare against a GMM output parametrized by Gaussians with either diagonal (MDN-Diag) or full (MDN-Full) covariance matrix. On the other hand, when assuming discrete support for the output distribution (i.e. we divide the map into tiled non-overlapping patches and view the pixels inside a patch as its measurements), we consider Convolutional LSTM (ConvLSTM), (Shi et al., 2015) which leverage the spatial information en-

---

[2]Please refer to the Appendix for the derivation.

*Table 1.* Test log-likelihood for each task under the continuous support assumption. For non-deterministic models the approximation on the marginal log-likelihood is given with the $\approx$ sign.

| Models | NYC-P | NYC-D | CPH-BS |
|---|---|---|---|
| RNN-MDN-Diag | 163582 | 143765 | 49124 |
| RNN-MDN-Full | 164016 | 146676 | 50109 |
| VRNN-MDN-Diag $\approx$ | 161345 | 139964 | 49231 |
| VRNN-MDN-Full $\approx$ | 162549 | 143671 | 49664 |
| SRNN-MDN-Diag $\approx$ | 164830 | 143719 | 49331 |
| SRNN-MDN-Full $\approx$ | 164976 | 147400 | 49810 |
| **RFN** $\approx$ | **168734** | **148291** | **51100** |

coded on the sequences by substituting the matrix operations in the standard LSTM formulation with convolutions.

**Results** *One-step Prediction*: In Table 1 we compare test log-likelihoods on the tasks of continuous spatio-temporal demand modelling for the cases of New York and Copenhagen. We report exact log-likelihoods for both RNN-MDN-Diag and RNN-MDN-Full, while in the case of VRNNs, SRNNs and RFNs we report the importance sampling approximation to the marginal log-likelihood using 30 samples, as in (Rezende et al., 2014). We see from Table 1 that RFN outperformes competing methods yielding higher log-likelihood across all tasks. The results support our claim that more flexible output distributions are advantageous when modelling potentially complex and structured temporal data distributions. To further illustrate this, in Fig. 2, we show a visualization of the predicted spatial densities (one-step-ahead) from three of the implemented models at specific times of the day. Opposed to GMM-based densities, the figures show how the RFN exploits the flexibility of conditional normalizing flows to generate sharper distributions capable of better approximating complex shapes such as geographical landforms or urban topologies (e.g. Central Park or the sharper edges in proximity of the Hudson river along the west side of Manhattan).

*Multi-step Prediction*: In order to take reliable strategic decisions, service providers might also be interested in obtaining full roll-outs of demand predictions, opposed to 1-step predictions. To do so, we generate entire sequences in an autoregressive way (i.e., the prediction at timestep $t$ is fed back into the model at $t + 1$) and analyze the ability of the proposed model to unroll for different forecasting horizons. From a methodological point of view, we are interested in measuring the effect of explicitly modelling the stochasticity in the temporal evolution of demand opposed to fully-deterministic architectures. To this regard, in Table 2 we compare the RFN with the most competitive deterministic benchmark (i.e. RNN-MDN-Full). As the results suggest, the stochasticity in the transition probability allows the RFN to better capture the temporal dynamics, thus resulting in lower performance decay in comparison with the

*Table 2.* Test log-likelihood comparison of RFN and RNN-MDN-Full for different forecast horizons on the NYC-P task.

| Models | t+2 | t+5 | t+10 | full (t+90) |
|---|---|---|---|---|
| RNN-MDN | 162891 | 161065 | 160099 | 158922 |
| **RFN** $\approx$ | **167509** | **167400** | **167359** | **167392** |

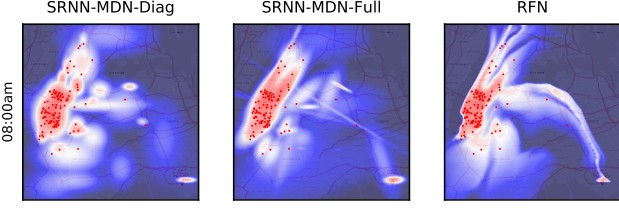

*Figure 2.* Generated spatio-temporal densities from SRNN-MDN-Diag, SRNN-MDN-Full and RFN on the NYC-P dataset. The blue (low) to red (high) log-likelihood heatmaps show models defined by increasing flexibility (best viewed in color).

fully deterministic RNN-MDN assuming full covariance.

*Quantization*: As a further analysis, we compare the proposed RFN with a Convolutional LSTM, under the assumption that the spatial map has been discretized in a $64 \times 64$ pixel space described by a Categorical distribution. This comparison is particularly relevant given the prevalence of ConvLSTMs in spatio-temporal travel modelling applications (Petersen et al., 2019; Yuan et al., 2018; Wang et al., 2018). As previously introduced, the RFN is naturally defined by a continuous output distribution (in practice parametrized as a normalizing flow), thus, in order to characterize a valid comparison, we apply a quantization procedure to obtain a discrete output distribution for the RFN. In particular, the implemented quantization procedure can be summarized with the following steps: (i) as in the continuous case, evaluate the approximated marginal log-likelihood under the trained RFN at the pixel-centers of a $64 \times 64$ grid, (ii) normalize the computed log-likelihood logits through the use of a softmax function and (iii) evaluate the log-likelihood under a Categorical distribution characterized by the probabilities computed in (ii), thus having values comparable with the output of the ConvLSTM. Table 3 compares test log-likelihoods on the task of discrete spatio-temporal demand modelling. To this regard, when considering results in Table 3, two relevant observations must be underlined. First of all, the true output of the RFN (i.e. before quantization) is a continuous density, thus, its discretization will, by definition, result in a loss of information and granularity. Secondly, and most importantly, the quantization is applied as post-processing evaluation step, thus, opposed to the implemented ConvLSTMs, the RFNs are not directly optimizing for the objective evaluated in Table 3. In light of this, the results under the discretized space assumption support even more our claims on the effectiveness of the RFN to approximate spatially complex

*Table 3.* Test log-likelihood for each task under the discrete support assumption. For the RFN, results are given after a quantization procedure mapping from a continuous 2d space to the $64 \times 64$ pixel space used to train the ConvLSTMs.

| Models | NYC-P | NYC-D | CPH-BS |
|---|---|---|---|
| ConvLSTM | -352962 | -350803 | -112548 |
| **RFN (Quantized)** | **-339745** | **-349627** | **-110999** |

distributions. Moreover, the ability of the RFN to model a continuous spatial density, opposed to a discretized approach as in the case of ConvLSTMs, has several theoretical and practical advantages. For instance, RFNs are be able to evaluate the log-likelihood of individual data points for anomaly and hotspot detection. Secondly, ConvLSTMs define a discretized space whose cells might have different natural landscape characteristics (e.g. rivers, lakes), thus effectively changing the dimension of the support in each bin and making comparisons of log-likelihoods across pixels an ill-posed question. Furthermore, if for discrete output distributions exploring different levels of discretization would require to repeatedly train independent ConvLSTM networks, the post-processing quantization of the RFN allows discretization to be done instantaneously, thus enabling for fast prototyping and exploration of discretization levels.

## 4. Related Work

A number of works have concentrated in defining more flexible emission functions for sequential models (Rasul et al., 2021; Kumar et al., 2020; Castrejon et al., 2019). Within this line of research, normalizing flows are used to either parametrize the emission function for the tasks of video generation and multi-variate time series forecasting (Castrejon et al., 2019; Rasul et al., 2021), or to describe the latent states representing the temporal evolution of the system (Kumar et al., 2020). However, to the author's best knowledge, there is no track of previous attempts for the task of urban mobility modelling.

Within the transportation domain, traditional approaches rely on spatial discretizations of the urban topology(Yuan et al., 2018; Petersen et al., 2019; Wang et al., 2018), which allow for the prediction of spatio-temporal sequences with discrete support through e.g. ConvLSTMs (Shi et al., 2015).

## 5. Conclusions

This work addresses the problem of continuous spatio-temporal density modelling by proposing the use of conditional normalizing flows as a general approach to parametrize the output distribution of recurrent latent variable models. Our experiments focus on real-world data for the task of urban mobility density modelling. We empirically show that the flexibility of normalizing flows enables

RFNs to generate rich output distributions capable of describing potentially complex geographical surfaces under both continuous and discrete output distribution assumptions. Ultimately, we believe that the ability to estimate fine-grained distributions of urban mobility represents an important step towards user-tailored MoD services.

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

# A. Appendix

## A.1. Training

We train each model using stochastic gradient ascent on the evidence lower bound $\mathcal{L}(\theta, \phi)$ defined in Eq. (7) using the Adam optimizer (Kingma & Ba, 2015), with a starting learning rate of $0.003$ being reduced by a factor of $0.1$ every $100$ epochs without loss improvement (in our implementation, we used the *ReduceLROnPlateau* scheduler in PyTorch with patience=100). As in (Sønderby et al., 2016), we found that annealing the KL term in Eq. (7) (using a scalar multiplier linearly increasing from 0 to 1 over the course of training) yielded better results. The final model was selected with an early-stopping procedure based on the validation performance. Training using a NVIDIA GeForce RTX 2080 Ti took around 6 hours for CPH-BS and around 9 hours for NYC-P/D.

## A.2. Benchmarks

For every model considered under the continuous support assumption, we select a single layer of 128 LSTM cells. The feature extractor $\varphi_\tau^{\text{extr}}$ in Eq. (1) has three layers of 128 hidden units using rectified linear activations (Nair & Hinton, 2010). For the VRNN, SRNN and RFN we also define a 128-dimensional latent state $\mathbf{z}_{1:T}$. Both the transition function $t_{\theta_z}$ from Eq. (2) and the inference network $\varphi_\tau^{\text{enc}}$ in Eq. (5) use a single layer of 128 hidden units. For the mixture-based models, the MDN emission is further defined by two layers of 64 hidden units where we use a softplus activation to ensure the positivity of the variance vector in the MDN-Diag case and a Cholesky decomposition of the full covariance matrix in MDN-Full. Based on a random search, we use 50 and 30 mixtures for MDN-Diag and MDN-Full respectively. The emission function in the RFN is defined as in Eq. (3) and Eq. (4), where $f_\psi$, $s_\psi$ and $t_\psi$ are neural networks with two layers of 128 hidden units. The conditional flow is further defined as an alternation of 35 layers of the triplet [Affine coupling layer, Batch Normalization (Ioffe & Szegedy, 2015), Permutation], where the permutation ensures that all dimensions are processed by the affine coupling layers and where the batch normalization ensures better propagation of the training signal, as shown in (Dinh et al., 2017). In our experiments we define $\mathbf{u}_t = \mathbf{x}_{t-1}$, although $\mathbf{u}_t$ could potentially be used to introduce relevant information for the problem at hand (e.g. weather or special event data in the case of spatio-temporal transportation demand estimation).

On the other hand, under the discrete support assumption, we train a 5-layer ConvLSTM network with 4 layers containing $40$ hidden states and $3 \times 3$ kernels (in alternation with $4$ batch normalization layers) using zero-padding to ensure preservation of tensor dimensions, a 3D Convolution layer with kernel $3 \times 3 \times 3$ and softmax activation function to describe a normalized density over the next frame (i.e. time-step) in the sequence.

All models assuming continuous output distribution were implemented using PyTorch (Paszke et al., 2017) and the universal probabilistic programming language Pyro (Bingham et al., 2018), while the ConvLSTMs where implemented using Tensorflow (Abadi et al., 2015). To reduce computational cost, we use a single sample to approximate the intractable expectations in the ELBO.

For both New York and Copenhagen experiments we process the data so to discard corrupted geo-coordinates outside the area of interest. For the taxi experiments, we discarded coordinates related to trips either shorter than $30s$ or longer than $3h$, while in the bike-sharing dataset, we ensured to keep only one app access from the same user in a window of $5$ minutes. In both cases we divide the data temporally into train/validation/test splits using a ratio of $0.5/0.25/0.25$.

## A.3. ELBO derivation

$$
\begin{aligned}
\log p_\theta(\mathbf{x}_{1:T}) &= \log \int p_\theta(\mathbf{x}_{1:T}, \mathbf{z}_{1:T}, \mathbf{h}_{1:T}) d\mathbf{z}\, d\mathbf{h} \\
&= \log \int \frac{q_\phi(\mathbf{z}_{1:T}|\mathbf{x}_{1:T})}{q_\phi(\mathbf{z}_{1:T}|\mathbf{x}_{1:T})} p_\theta(\mathbf{x}_{1:T}, \mathbf{z}_{1:T}, \mathbf{h}_{1:T}) d\mathbf{z}\, d\mathbf{h} \\
&= \log \mathbb{E}_{q_\phi(\mathbf{z}_{1:T}|\mathbf{x}_{1:T})} \left[ \prod_{t=1}^{T} \frac{p_\theta(\mathbf{x}_t|\mathbf{z}_t, \mathbf{h}_t) p_\theta(\mathbf{z}_t|\mathbf{z}_{t-1}, \mathbf{h}_t) p_\theta(\mathbf{h}_t|\mathbf{h}_{t-1}, \mathbf{u}_t)}{q_\phi(\mathbf{z}_t|\mathbf{z}_{t-1}, \mathbf{h}_t, \mathbf{x}_t)} \right] \\
&\geq \mathbb{E}_{q_\phi(\mathbf{z}_{1:T}|\mathbf{x}_{1:T})} \left[ \sum_{t=1}^{T} \log p_\theta(\mathbf{x}_t|\mathbf{z}_t, \mathbf{h}_t) + \log p_\theta(\mathbf{h}_t|\mathbf{h}_{t-1}, \mathbf{u}_t) + \log \left( \frac{p_\theta(\mathbf{z}_t|\mathbf{z}_{t-1}, \mathbf{h}_t)}{q_\phi(\mathbf{z}_t|\mathbf{z}_{t-1}, \mathbf{h}_t, \mathbf{x}_t)} \right) \right] \qquad (7) \\
&= \mathbb{E}_{q_\phi(\mathbf{z}_{1:T}|\mathbf{x}_{1:T})} \left[ \sum_{t=1}^{T} \log p_\theta(\mathbf{x}_t|\mathbf{z}_t, \mathbf{h}_t) + \log p_\theta(\mathbf{h}_t|\mathbf{h}_{t-1}, \mathbf{u}_t) \right] \\
&\quad - \sum_{t=1}^{T} \mathbb{KL} \left( q_\phi(\mathbf{z}_t|\mathbf{z}_{t-1}, \mathbf{h}_t, \mathbf{x}_t) || p_\theta(\mathbf{z}_t|\mathbf{z}_{t-1}, \mathbf{h}_t) \right) = \mathcal{L}(\theta, \phi)
\end{aligned}
$$