# OpenReview forum: "Recurrent Flow Networks: A Recurrent Latent Variable Model for Density Modelling of Urban Mobility"
_ICML.cc/2021/Workshop/INNF — INNF+ 2021 poster_

### Official Review · Reviewer_Xt1J · 2021-06-10

**Rating:** Borderline Reject
**Confidence:** 4

**Summary:**

The authors are interested in modelling temporal data. Their target application is the modelling of Mobility-on-Demand systems, such as Taxi services, or Bike sharing data. Their motivation is to create more flexible models that can capture better complex data. Their proposed model is a recurrent graphical model. The model is trained using an Evidence Lower Bound, as in VAEs and State Space Models, but it also contains some Flow-like layers, with an affine coupling layer.

**Justification For Rating:**

The definition of the model in Fig 1, and in equations 3 and 4 is confusing. For example:
* it's not clear what is the role of random variable 'b'? What motivates the addition of this variable?
* Eq 4 can obviously be inverted to define x as a function of b, but this equation is written the other way round with b as a function of x.
* Why does Eq 4 look like a single coupling layer that copies b_{t,1:d} onto x_{t,1:d}, instead of stacking coupling layers, alternating between frozen variables, as is usually done with normalizing flows.

All these means that I did not feel like I could understand the model, and the motivation behind its structure.

Other small comments:
* Term SRNN is first used on line 104, but only defined on line 146.
* In equation (4), the variables 'd' and 'D' are introduced without being defined.
* On line 147, the term MDN is used without being defined. This term is re-used many times, still without being defined.
* One of the test dataset is made of bike sharing data. Since this is a very valid application, it would be good if the introduction mentioned it as well (at the moment, it only mentions car rental systems).

---

### Official Review · Reviewer_3att · 2021-06-10

**Rating:** Borderline Accept
**Confidence:** 3

**Summary:**

This paper introduces Recurrent Flow Networks (RFNs), an approach that combines stochastic and deterministic temporal hidden variables with conditional normalizing flows (NFs) for modelling temporal sequences. Taking inspiration from Dynamic Bayesian Networks and RNNs the proposed approach combines deterministic with stochastic temporal representations. Then a conditional NF is used to model the conditional distribution of the observation given the temporal representation. Experiments show the model outperforms other methods on urban mobility datasets.

**Justification For Rating:**

The proposed model combined exact likelihood models with variational training. I think details about the training procedure are lacking. Which part of the model (if not all) is responsible for the improvement with respect to other methods is not clear. Doing an analysis of this would be interesting.
Overall the idea of combining deterministic with stochastic features in this context may be interesting however it is not clear why then using a normalizing flow is useful for modelling the conditional distribution (as the stochastic temporal encoding could embed the required multimodality).
Figure 1 is very confusing as well.
Overall the motivation for introducing such complexity in the model is not very clear which weaken the message of the paper.

---

### Official Review · Reviewer_DGCD · 2021-06-14

**Rating:** Borderline Reject
**Confidence:** 3

**Summary:**

This works combines a sequential latent variable model with conditional normalizing flow decoder. The target application is urban mobility and the data sets contain sequences of geo-coordinates (2D).

**Justification For Rating:**

The paper is on topic, but it is a bit confusing. From what I understand, the model is a sequential latent variable model with a conditional normalizing flow decoder, and the data sets are sequences of geo-locations (2D) related to urban mobility activities.

What I think the paper could have done better:

1. Model ablation / compositionally. The use of both latent variables and a complex decoder are often not necessary. Does z provide any usefulness when the decoder is a normalizing flow? It seems with a strong enough flow model, this could have been an exact likelihood model, with no latent variables. Perhaps the combination of both can lead to smaller models, in which case a parameter count table could be helpful.

2. Notation / graphical models. We usually wouldn't put deterministic states into conditional probabilities. Just some examples: the hidden state is deterministic, so p(h_t | h_{t-1}, u_t) is a Dirac delta. Also, p(z_t | z_{t-1}, h_t) should likely be p(z_t | z_{t-1}, u_t). The former marginalizes out all u_t that map to the value of h_t, which isn't quite right. In Figure 1, if b_t and x_t are bijections, only one needs to be shown, usually x_t because b_t is only used to define the distribution over x_t.

3. Car/bike pickups/dropoffs occur in irregular intervals, and I'm not sure how this was preprocessed into a discrete-time sequence. Is the model tasked with predicting the next geo-location regardless of how much time has elapsed from the previous observation? A model that takes into account irregular sampling would be more ideal.

4. There should be multiple runs for each model and report standard deviations. The test log-likelihoods are very close to each other. Also, these should be negative log-likelihood values, right?

I think the quantization section is interesting. It's not often enough that papers compare to simple baselines like a discrete-space model.

---

### Decision · Program_Chairs · 2021-06-14

**Decision:**

Accept (poster)

**Comment:**

The paper is on-topic for the workshop, and considers an interesting application. However, the reviewers found that parts of the model are not clear, the model design could be better motivated, and the individual model choices could be better evaluated experimentally. Although borderline, we decided to accept the paper, in the hope that these issues can be discussed and clarified during the workshop.